# Compressive Strength, Chloride Ion Penetrability, and Carbonation Characteristic of Concrete with Mixed Slag Aggregate

**DOI:** 10.3390/ma13040940

**Published:** 2020-02-20

**Authors:** Se-Jin Choi, Young-Uk Kim, Tae-Gue Oh, Bong-Suk Cho

**Affiliations:** 1Department of Architectural Engineering, Wonkwang University, 460 Iksan-daero, Iksan 54538, Korea; alkjd3@naver.com; 2Department of Architectural Engineering, Yonsei University, 50 Yonsei-ro Seodaemun-gu, Seoul 03722, Korea; yuk92@hanmail.net; 3Environment and Resources Research Group, Research Institute of Industrial Science and Technology, Pohang 37673, Korea; chos8@hanmail.net

**Keywords:** aggregate, blast furnace slag, ferronickel slag, compressive strength, chloride ion penetrability, carbonation

## Abstract

The shortage of natural aggregates has recently emerged as a serious problem owing to the tremendous growth of the concrete industry. Consequently, the social interest in identifying aggregate materials as alternatives to natural aggregates has increased. In South Korea’s growing steel industry, a large amount of steel slag is generated and discarded every year, thereby causing environmental pollution. In previous studies, steel slag, such as blast furnace slag (BFS), has been used as substitutes for concrete aggregates; however, few studies have been conducted on concrete containing both BFS and Ferronickel slag (FNS) as the fine aggregate. In this study, the compressive strength, chloride ion penetrability, and carbonation characteristic of concrete with both FNS and BFS were investigated. The mixed slag fine aggregate (MSFA) was used to replace 0, 25%, 50%, 75%, and 100% of the natural fine aggregate volume. From the test results, the highest compressive strength after 56 days was observed for the B/F100 sample. The 56 days chloride ion penetrability of the B/F75, and B/F100 samples with the MSFA contents of 75% and 100% were low level, approximately 34%, and 54% lower than that of the plain sample, respectively. In addition, the carbonation depth of the samples decreased with the increase in replacement ratio of MSFA.

## 1. Introduction

Owing to the immense growth of the global concrete industry, the shortage of natural aggregates has emerged as a serious problem. In Korea, the lack of aggregates has often led to construction problems. Therefore, a considerable amount of social and research interest has been focused on finding alternative aggregate materials to replace natural aggregates [1,2,3,4,5,6,7]. Various types of steel slag can be considered as alternatives to aggregates for concrete. Ferronickel slag (FNS) is an industrial byproduct of the ferronickel production process. It is obtained after nickel ore and bituminous coal used as raw materials in the ferronickel smelting process are melted at a high temperature and separated from ferronickel [2]. The annual amount of FNS produced in South Korea is over 2 million tons. Most are discarded and cause serious environmental pollution. Generally, the FNS is used as a substitute material for foundry sand, abrasive, and serpentine [2]. In addition, studies were carried out on the use of FNS as fine aggregate for concrete [5,6,7].

Saha et al. [5] studied the strength and durability of cement mortar using FNS as the replacement of natural sand. The maximum compressive strength of cement mortar was obtained by replacing 50% sand with FNS. In addition, X-ray diffraction test results showed that the pozzolanic reaction of fly ash helped to reduce the strength loss.

Choi et al. [6] investigated the alkali-silica reactivity of cementitious materials with FNS fine aggregate produced under different cooling conditions. The alkali-silica reactivity of mortar using FNS fine aggregate was dependent on the cooling speed and particle size of FNS.

Lee et al. [7] investigated the mechanical properties and resistances to freezing and thawing of concrete using an air-cooled ferronickel slag (ACFNS) fine aggregate. The compressive strength and static modulus of elasticity of the concrete with ACFNS fine-aggregate increased with increasing the replacement ratio of ACFNS.

Blast furnace slag (BFS) is also another steel industry byproduct that is obtained from blast furnaces used in the manufacturing of pig iron. The annual amount of produced BFS in Korea is approximately 15 million tons. The BFS has been extensively used as a successful replacement material for Portland cement in concrete materials to improve the durability and in the production of high-strength concrete, with environmental and economic benefits, such as resource conservation, CO_2_ reduction, and energy savings [8,9,10,11]. In addition, BFS also can be used as the aggregate for cement mortar of concrete. In previous studies [12,13], BFS was used as a substitute material for concrete aggregate; however, few studies have been conducted on concrete with slag aggregate containing both FNS and BFS as the fine aggregate.

In this study, the slump, air content, compressive strength, resistance to chloride ions, and carbonation characteristic of concrete with both FNS and BFS as the fine aggregate were investigated to effectively utilize the mixed slag fine aggregate (MSFA) as a substitute material for the natural aggregate in the concrete industry.

## 2. Materials and Methods

### 2.1. Materials

An ASTM type I ordinary Portland cement manufactured by Asia Cement Co. (Seoul, Korea), BFS powder obtained from Daehan Slag Co., Ltd. (Gwangyang, Korea), and fly ash obtained from the Honam power plant in Korea were used as cementitious materials in this study. In addition, a crushed coarse aggregate (Granite, G_max_ 25 mm) with a density of 2.65 and fineness modulus of 6.49 was used.

Table 1 summarizes the chemical compositions of the cement, BFS powder, and fly ash used in the experiment. Natural, BFS, and FNS fine aggregates were used in the experiment. Natural sand (NS) was used as the natural fine aggregate with a maximum size of 5 mm and fineness modulus of 2.89. The BFS sand (BS) and FNS sand (FS) used as the slag fine aggregates were obtained from POSCO, Korea.

Figure 1 shows the used fine aggregate samples, while Table 2 summarizes their physical properties. Figure 2 shows the particle size distributions of the NS, BS, FS, and MSFA (B/F) with a BFS:FNS ratio of 5:5 by volume. The particle size distribution of each aggregate was compared with the standard proposed by KS F 2527. The fineness modulus of the BS was smaller than that of the NS, while that of the FS was higher than that of the NS. The fineness modulus of B/F was 2.94, similar to that of the NS.

### 2.2. Mixing Proportions and Specimen Preparation

In this study, the MSFA with the BFS:FNS mixture ratio of 5:5 was used to replace 0 (plain), 25%, 50%, 75%, and 100% of the volume of the NS. A constant water-to-binder ratio of 0.518 was used. In all mixtures, the BFS powder and fly ash were used to replace 20% and 10% (weight) of the cement, respectively. The mixing proportions of the concrete samples are summarized in Table 3. In addition, a water-reducing agent (WRA; S Co., Seoul, Korea) was used to control the fluidities of all mixtures. The components of the concrete samples were mixed in a mechanical mixer. Cylindrical molds (∅100 × 200 mm) were fabricated for the compressive strength test. After 24 h, the specimens were removed from their molds and cured at 20 °C in a water tank.

The slump and air content tests of the concrete samples were carried out in accordance with Korean Standards (KS) F 2402 [14] and KS F 2421 [15], respectively. The compressive strength test was carried out after 7, 14, 28, and 56 days in accordance with KS F 2405 [16]. The presented strength test values are the average values of three samples.

Chloride ion penetration tests were carried out after 7, 14, 28, and 56 days, according to ASTM 1202 C [17]. Specimens having dimensions of ∅100 × 50 mm, obtained by cutting the ∅100 × 200 mm cylindrical specimens, were used in the test. The specimen and equipment used in the chloride ion penetration test are shown in Figure 3.

Accelerated-carbonation test (Figure 4) of the concrete samples (∅100 × 200 mm) was carried out during 7, 28, and 56 days, according to KS F 2584 [18], by using an accelerated-carbonation chamber at a constant temperature of 20 ± 2 °C, constant humidity of 60 ± 5%, and constant CO_2_ concentration of 5 ± 0.2%. During the testing period, the samples were split into two halves and the carbonation depth was measured by spraying an approximately 1% phenolphthalein solution on the broken surface of the sample, after the dust was removed.

## 3. Results and Discussion

### 3.1. Slump and Air Content

Figure 5 shows the slumps and WRA dosages of the samples with the MSFAs. The slumps of all mixtures were similar, in the range of 200 to 210 mm, regardless of the replacement ratio of MSFA. In addition, the dosage of WRA used to control the fluidity of the plain sample with the NS was 0.9% of the binder weight. The WRA dosage decreased with the increase in replacement ratio of MSFA. The WRA dosage of the B/F100 sample (only with the MSFA) was 0.2% of the binder weight. The tendency that the fluidity of the mixture with the BFS fine aggregate is better than that of the mixture with the NS owing to the vitreous texture of the BFS particle is similar to those in previous reports [13,19].

Figure 6 shows the variation in air content of the concrete sample with both BFS and FNS as the fine aggregate with the replacement ratio of MSFA. The air contents of the concrete samples were similar (2.3% to 2.6%), regardless of the replacement ratio of MSFA.

### 3.2. Compressive Strength

Figure 7 shows the variation in compressive strength of the concrete sample with both BFS and FNS as the fine aggregate with the replacement ratio of MSFA. After seven days, the compressive strength of the plain sample without MSFA was approximately 23.1 MPa, while those of the samples with the MSFAs were in the range of 21.2 to 23.5 MPa. After 14 days of curing, the compressive strengths of all samples, except the B/F75 sample, were similar (approximately 29 MPa). After 28 days of curing, the compressive strength of the plain sample was approximately 33.5 MPa, while those of the samples with the MSFAs were in the range of 32.2 to 34.3 MPa. The compressive strength of the sample with the MSFA increased with the replacement ratio of MSFA. After 56 days, the compressive strengths of all samples were increased; those of the samples with the MSFAs were in the range of 36.6 to 38.8 MPa. The highest compressive strength (approximately 38.8 MPa) was obtained for the B/F100 sample, which contained only the MSFA. The increase in compressive strength could be explained as the particle size distribution of the MSFA was similar to that of the NS and the formation of a secondary calcium silicate hydrated (CSH) gel was initiated [20].

### 3.3. Chloride Ion Penetrability

Figure 8 shows the variation in chloride ion penetrability of the sample with both BFS and FNS as the fine aggregate. The total charge passed through the sample during the considered period was calculated according to ASTM C 1202. After seven days, the charge passed through the plain sample was approximately 9273 C. The charge passed through the B/F100 sample with 100% MSFA was the smallest, approximately 37% smaller than that of the plain sample. After 14 days of curing, the largest passed charge (approximately 8083 C) was observed for the plain sample, which contained only the NS. The charges passed through all samples with the MSFAs were smaller than that through the plain sample. The charge passed through the B/F100 sample was the smallest (4106 C), approximately 50% smaller than that through the plain sample (8084 C). After 28 days of curing, the charge passed through the sample decreased with the increase in replacement ratio of MSFA (3993 C (plain) to 2041 C (B/F100)). The chloride ion penetrabilities of all samples were moderate level (2000–4000 C; ASTM C 1202 [17]). After 56 days, the charge passed through the sample decreased with the increase in replacement ratio of MSFA. The chloride ion penetrabilities of B/F50, B/F75, and B/F100 with MSFA contents of 50%, 75%, and 100% were low level (1000–2000 C; ASTM C 1202), approximately 17%, 34%, and 54% lower than that of the plain sample, respectively. The resistances to penetration of chloride ions of the samples with the MSFAs were better than that of the plain sample. The tendency that the concrete with BFS has a good resistance to chloride ions is similar to those observed in previous studies [21,22].

Figure 9 shows the relation between the compressive strength and chloride ion penetrability for the samples with different replacement ratios of MSFA. The chloride ion penetrability decreased with the increase in compressive strength. In addition, the chloride ion penetrabilities of the samples with the MSFAs were lower than that of the sample with the NS at the same compressive strength.

### 3.4. Carbonation Depth

Figure 10 shows the variation in carbonation depth of the sample with both BFS and FNS as the fine aggregate. A higher replacement ratio of MSFA led to a smaller carbonation depth. After seven days of treatment in the accelerated carbonation chamber, the carbonation depth of the plain sample was approximately 1.18 mm, while those of the samples with the MSFAs were approximately 45% to 69% (0.53 to 0.82 mm) of that of the plain sample. After 28 days, the carbonation depths of the plain, B/F25, B/F50, B/F75, and B/F100 samples were approximately 1.16, 0.94, 0.83, 0.76, and 0.68 mm, respectively. The carbonation depth of B/F100 was approximately 41% smaller than that of the plain sample. After 56 days of accelerated carbonation testing, the carbonation depths of all samples were increased. The largest carbonation depth (1.26 mm) was observed for the plain sample. The carbonation depth decreased with the increase in replacement ratio of MSFA. The carbonation depth of B/F100 was the smallest (0.79 mm). The tendency that the resistance to carbonation of the concrete with the steel slag as the aggregate is better than that of the concrete with the natural aggregate is similar to that in a previous report [23]. This shows that the use of the MSFA in the mortar or concrete can be effective for the improvement in carbonation resistance.

Figure 11 shows the relation between the compressive strength and carbonation depth for the concrete samples with different replacement ratios of MSFA. With the increase in accelerated carbonation testing period, the carbonation depth increase was accompanied by an increase in compressive strength. In addition, the carbonation depths of the samples with the MSFAs were smaller than that of the plain sample at the same compressive strength.

## 4. Conclusions

The conclusions of this study can be summarized as follows.
(1)The slumps of all mixtures were similar (200 to 210 mm), regardless of the replacement ratio of MSFA. The WRA dosage decreased with the increase in replacement ratio of MSFA.(2)The compressive strength of the plain sample was approximately 23.9 MPa, while those of the samples with the MSFAs were in the range of 21.2 to 23.5 MPa after seven days. After 56 days, the highest compressive strength (approximately 38.8 MPa) was observed for the B/F100 sample. The increase in compressive strength could be explained as the particle size distribution of the MSFA was similar to that of the NS and the formation of the secondary CSH gel was initiated.(3)After seven days, the charge passed through B/F100 was the smallest, approximately 37% smaller than that through the plain sample. After 28 days of curing, the chloride ion penetrabilities of all samples were moderate level according to ASTM C 1202. After 56 days, the chloride ion penetrabilities of B/F50, B/F75, and B/F100 were low level, approximately 17%, 34%, and 54% lower than that of the plain sample, respectively.(4)The resistances to penetration of chloride ions of the samples with the MSFAs were better than that of the plain sample. The tendency that the concrete with BFS has a good resistance to chloride ions is similar to those in previous reports.(5)The chloride ion penetrability decreased with the increase in compressive strength. In addition, the chloride ion penetrabilities of the samples with the MSFAs were lower than that of the plain sample at the same compressive strength.(6)The higher replacement ratio of MSFA led to a smaller carbonation depth. The carbonation depth (0.79 mm) of B/F100 was the smallest after 56 days. The results show that the use of the MSFA in the mortar or concrete can be effective for the improvement in carbonation resistance.

However, further studies are needed to establish the strength development mechanism and respective relationships between the strength properties of mortar containing various slag aggregate and water-binder ratio, density, alkali content, and durability, etc.

## Figures and Tables

**Figure 1 materials-13-00940-f001:**
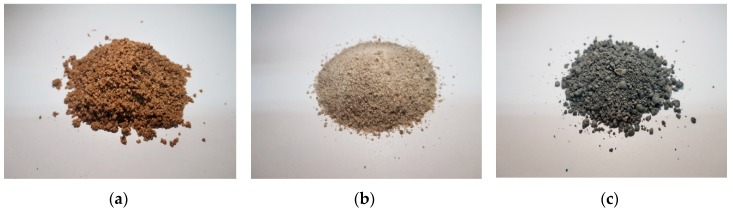
Fine aggregate samples. (**a**) Natural sand (NS); (**b**) BFS sand (BS); (**c**) FNS sand (FS).

**Figure 2 materials-13-00940-f002:**
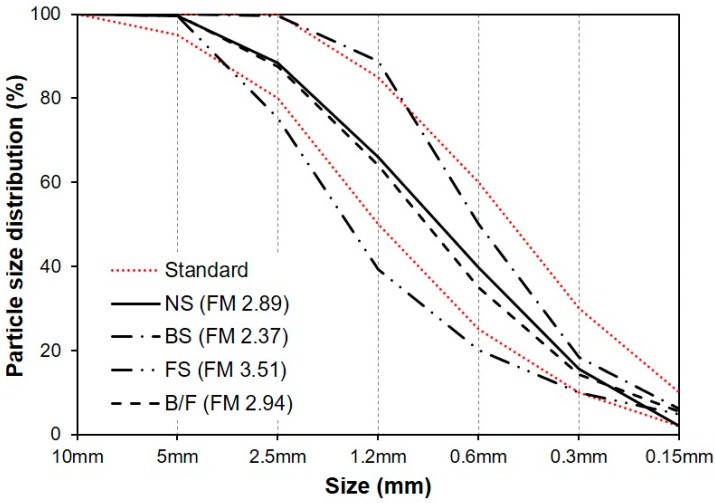
Particle size distribution of fine aggregate.

**Figure 3 materials-13-00940-f003:**
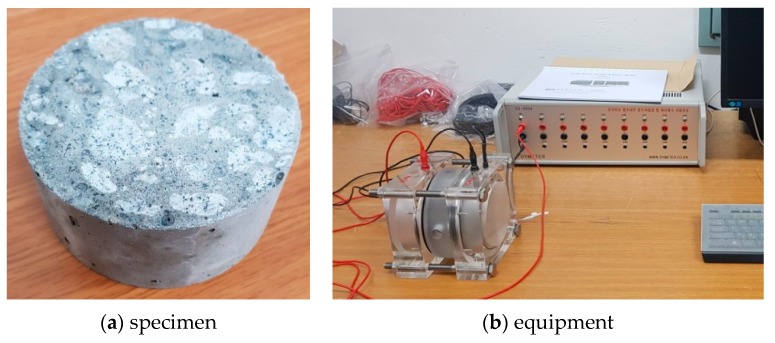
Specimen and equipment for chloride ion penetration test.

**Figure 4 materials-13-00940-f004:**
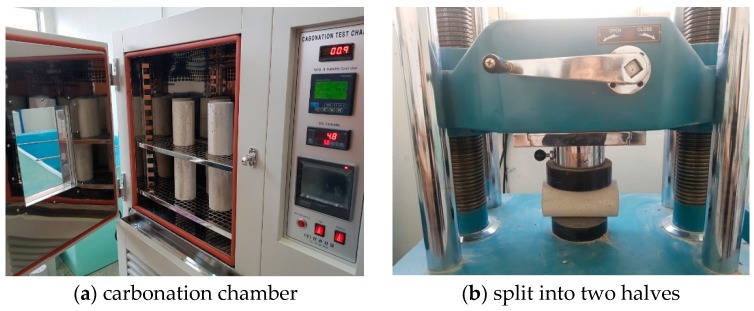
Accelerated carbonation test.

**Figure 5 materials-13-00940-f005:**
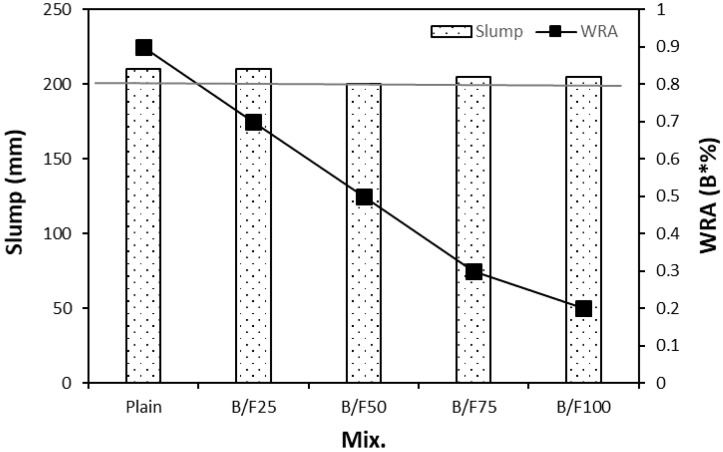
Slump and WRA dosage.

**Figure 6 materials-13-00940-f006:**
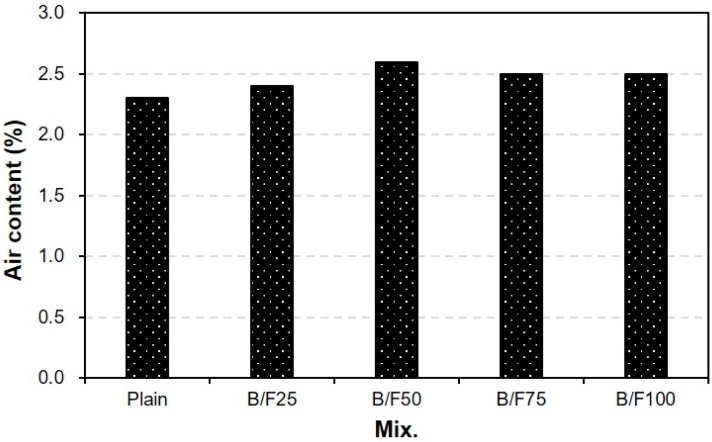
Air content.

**Figure 7 materials-13-00940-f007:**
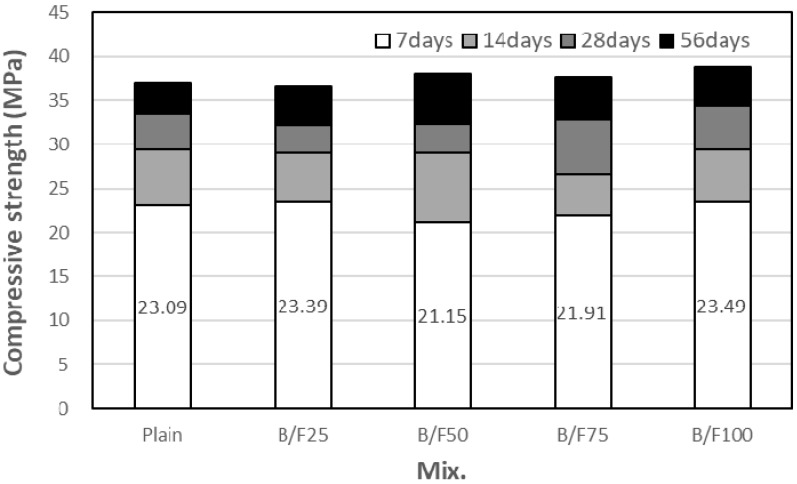
Compressive strength.

**Figure 8 materials-13-00940-f008:**
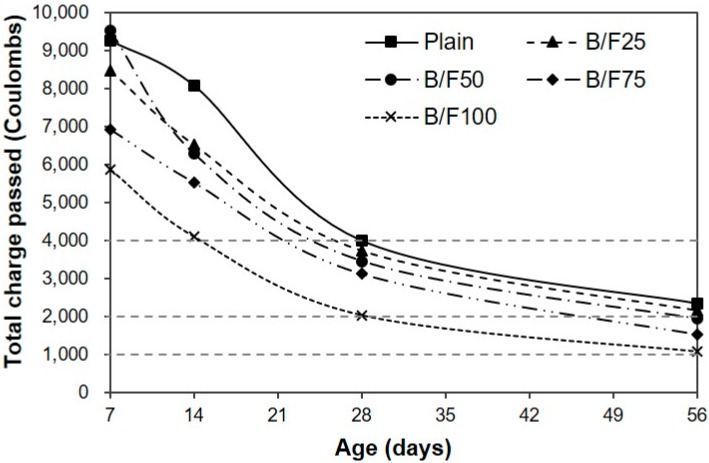
Chloride ion penetrability.

**Figure 9 materials-13-00940-f009:**
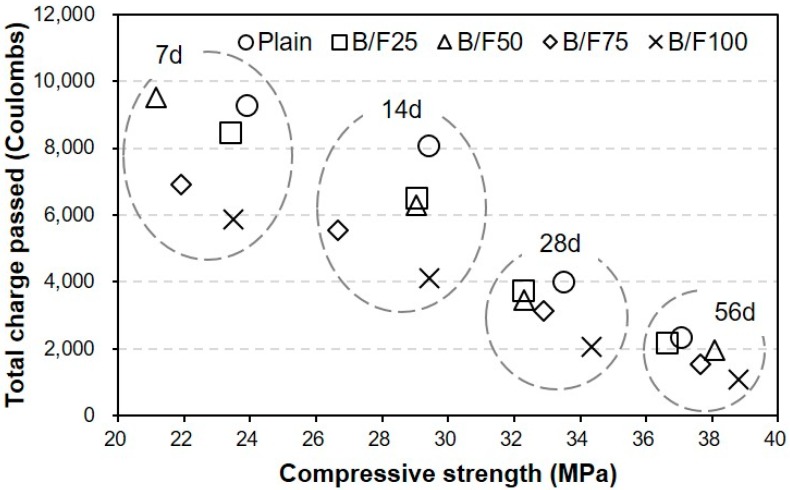
Relationship between compressive strength and chloride ion penetrability.

**Figure 10 materials-13-00940-f010:**
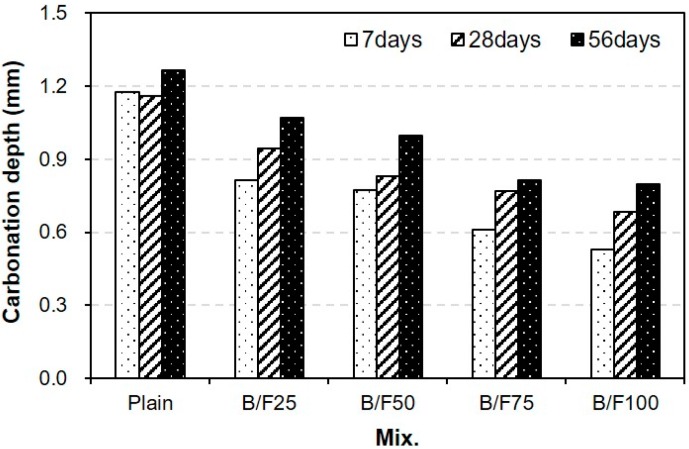
Carbonation depth.

**Figure 11 materials-13-00940-f011:**
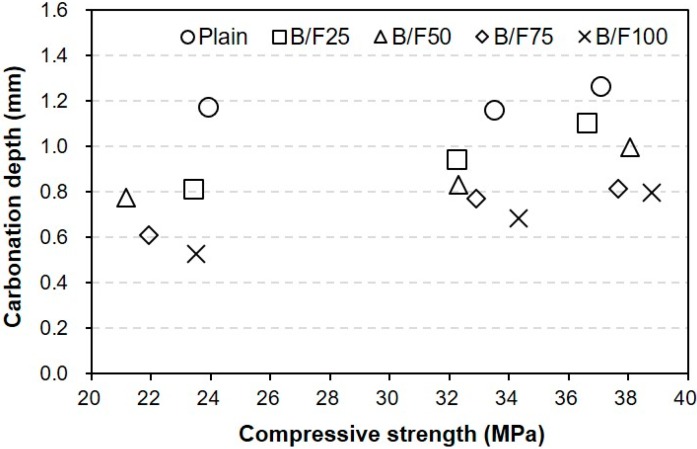
Relationship between compressive strength and carbonation depth.

**Table 1 materials-13-00940-t001:** Chemical composition and strength of cementitious materials.

Type	SiO_2_	Al_2_O_3_	Fe_2_O_3_	CaO	MgO	K_2_O	Comp. str. (MPa)	
							7 day	28 day
Cement (C)	17.43	6.50	3.57	64.40	2.55	1.17	42.7	56.5
Blast furnace slag (BFS) powder	30.61	13.98	0.32	40.71	6.43	0.60	-	-
Fly ash (FA)	64.88	20.56	6.06	2.58	0.80	1.45	-	-

**Table 2 materials-13-00940-t002:** Physical properties of fine aggregates.

Type	FM	Density (g/cm^3^)	Water Absorption (%)	Unit Weight (kg/L)	Ratio of Absolute Volume (%)
Natural sand (NS)	2.89	2.63	1.1	1.645	62.56
BFS sand (BS)	2.37	2.81	2.1	1.737	61.80
FNS sand (FS)	3.51	3.04	0.6	1.871	61.56

**Table 3 materials-13-00940-t003:** Mix proportion of concrete.

Mix	W/B (%)	S/a (%)	Unit Weight (kg/m^3^)								WRA (B*%)
			Water	Cement	BFS Powder	Fly Ash	NS	BS	FS	Gravel	
Plain	51.8	47	176	238	68	34	812	-	-	916	0.9
B/F25			176	238	68	34	609	109	117	916	0.7
B/F50			176	238	68	34	406	219	234	916	0.5
B/F75			176	238	68	34	203	328	351	916	0.3
B/F100			176	238	68	34	-	437	469	916	0.2

B*: Binder.

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
