# Peer review of "Compressive Strength, Chloride Ion Penetrability, and Carbonation Characteristic of Concrete with Mixed Slag Aggregate"

_materials, 2020, doi:10.3390/ma13040940_

Round 1

Reviewer 1 Report

The introduction is non-exhaustive and sketchy.

line 66 - what strength class cement was used?

There is no information on sample density.

Please explain all the signs in Table 3. The nomenclature is missing.

In figures 8 and 9 on the ordinate axes, reduce the number of decimal places. Use dots to write numbers.

In the graphs (Figure 5, 6, 7, 10) there is no marked dispersion of the results obtained.

The article presents only the results of research and obtained dependencies without further reflection on the reason for their physical justification.

Author Response

Thank you for your comments to the manuscript.

Authors have revised the manuscript as below in accordance with reviewer’s comments.

We would like to resubmit this paper, and we hope you will consider this paper as suitable for publication in your journal.

We are looking forward to your reply.

Thank you in advance.

Reviewer 2 Report

Line 68 - coarse aggregate: type of rock and maximum particle size to be stated

Line 71 - Table 1: total equivalent alkali content in relation to study in Ref. 6 if known

Line 91 - SFS:FNS ratio of 5:5: by volume (?)

Line 96 - Figure 2: Standard of grading limits to be provided

Line 186 - dosage of WRA: besides grading (FM) and surface texture, particle shape is also a factor.  Magnified view of aggregates will be useful, if available.

Line 209 - increase in strength: trend is clear but actual percentages are uncertain statistically (number of specimen for mean and estimated standard deviation may indicate confidence of differences around 3 MPa). 

Author Response

(The authors gave the same response as above.)
